# Methodologies for Monitoring the Digital Marketing of Foods and Beverages Aimed at Infants, Children, and Adolescents (ICA): A Scoping Review

**DOI:** 10.3390/ijerph19158951

**Published:** 2022-07-23

**Authors:** Vania Lara-Mejía, Bianca Franco-Lares, Ana Lilia Lozada-Tequeanes, Casandra Villanueva-Vázquez, Sonia Hernández-Cordero

**Affiliations:** 1Center for Equitable Development EQUIDE, Universidad Iberoamericana Ciudad de México, Prolongación Paseo de la Reforma 880, Col. Lomas de Santa Fe, Álvaro Obregón, Mexico City 01219, Mexico; v.lara.m.16@gmail.com (V.L.-M.); lnbiancafranco@gmail.com (B.F.-L.); cassandravillanueva14@hotmail.com (C.V.-V.); 2Research Center of Nutrition and Health, National Institute of Public Health, Av. Universidad 655, Col. Sta. Ma. Ahuacatitlán, Cuernavaca 62100, Mexico; cinys20@insp.mx

**Keywords:** scoping review, marketing, digital marketing, social media, breast-milk substitutes, high-fat foods, high-sugar beverages, infant nutrition, child nutrition, adolescent nutrition

## Abstract

While television has been the most widely used medium for food and beverage marketing, companies are shifting in favor of digital media. The ubiquitous digital marketing of breast-milk substitutes (BMS) and foods and beverages high in saturated fat, salt, and/or free sugars (FBHFSS) has been considered a powerful environmental determinant of inadequate dietary practices during infancy, childhood, and adolescence. The scoping review’s aim was to systematically identify and map the types of methodologies available to monitor the digital marketing of foods and beverages targeting infants, children, and adolescents (ICA) worldwide. Research evidence published from 2011 to October 2021 was examined using search strategies including multiple databases and citation tracking. A total of 420 sources were evaluated, and 28 studies from 81 countries meeting the inclusion criteria were retained. Most of the studies (*n* = 24) documenting methodologies to monitor inappropriate digital marketing were published since 2015 and were primarily aimed at identifying the promotional techniques and nutritional content of FBHFSS targeting adolescents (*n* = 13). It is paramount to develop a feasible and scalable monitoring system to develop effective policies to protect parents and ICA from BMS and FBHSFF digital marketing.

## 1. Introduction

In recent decades, public health experts on infant and young child nutrition (IYCN) have tended to focus on the importance of breastfeeding and healthy eating habits in the first years of life as one of the most effective ways to ensure the development and health of infants, children, and adolescents (ICA) [1]. The World Health Organization (WHO) Report of the Commission on Ending Childhood Obesity recommends promoting the health benefits of a nutritionally sound diet throughout a life-course by investing in nourishment from the earliest stages of life [2]. During infancy and early childhood, this could be achieved through the protection, promotion, and support of breastfeeding and an appropriate complementary feeding. Later, during childhood and adolescence, by promoting and enhancing healthy eating in the family and school environment. The ubiquitous marketing of foods and beverages high in saturated fat, salt, and/or free sugars (FBHFSS) has been associated with inadequate feeding practices and is hence considered a powerful environmental determinant of childhood and adolescent obesity and other chronic diseases (e.g., diabetes and high blood pressure, among others) [3]. As a result, national and international health organizations have called on governments to adopt policies to limit FBHFSS promotion [4]. Furthermore, for younger infants, mothers’ decisions on infant feeding practices are influenced by individual, socio-economic, cultural, and environmental factors, including the pervasive marketing of breast-milk substitutes (BMS) manufacturers [5]. BMS marketing has contributed negatively to optimal infant feeding practices, decreasing breastfeeding rates [6,7].

The WHO and the United Nations International Children’s Emergency Fund (UNICEF), through official recommendations, explicitly call for global action to reduce the impact on ICA of the marketing of BMS and FBHFSS in all media by introducing general restrictions. Firstly, in order to restrict marketing practices that influence parents’ feeding decisions for infants and young children, in 1981, the 34th World Health Assembly (WHA) adopted the International Code of Marketing of Breast-milk Substitutes (subsequently referred to as ‘the Code’) [8]. It aimed to control the inappropriate marketing of BMS to protect and promote breastfeeding and to ensure the appropriate use of BMS, when these are necessary [9]. According to the Code, BMS are all foods and beverages marketed for feeding infants and young children up to three years of age, including infant and follow-up formula, other milk products, bottle-feed complementary foods, feeding bottles, and teats. Since then, the WHA has adopted several subsequent relevant resolutions to clarify and strengthen aspects of the Code [10]. Up to March 2022, 144 (74.2%) of 194 WHO Member States have adopted legal measures to regulate BMS marketing aligned with the Code [11]. Of these, only 37 explicitly mention the promotion of BMS on the internet, digital channels, or other electronic media [11]. Similarly, to ensure that ICA around the world are protected against the impact of FBHFSS marketing, and given the opportunity to grow and develop in an enabling food environment, in 2010, the WHO proposed a set of recommendations. These recommendations are intended to guide Member States’ efforts in designing new and/or strengthening existing policies on FBHFSS marketing practices [4].

Currently, both the BMS and FBHFSS industries have strengthened and expanded their marketing strategies in a more sophisticated and specialized way [12]. Although television has been the most widely used medium for FBHFSS marketing, companies are shifting in favor of digital media [13]. This is due to the growing role of digital media in human lives and the great potential for marketers to reach this audience [14]. The use of novel marketing techniques (e.g., advergaming games that promote a particular brand or product by integrating it into play) in combination with paid media (e.g., pop-up advertisements), owned media (e.g., branded websites and social media pages), and content co-created with users have improved the scope of display ads [14]. Digital media allow marketers to contact parents, children, and adolescents in real time to send unprecedented volumes of information, often using Artificial Intelligence (AI)-enabled tactics [15,16]. As the industry has leveraged AI to market FBHFSS in digital media, monitoring has been difficult because of the volume (e.g., millions of websites), velocity (e.g., continuously changing content), and variety (e.g., games, display adverts) of digital marketing. Similarly, initiating consumer defenses is more difficult because digital marketing is harder to recognize since commercial messages are intimately integrated with content and no longer interrupt, like those on television [12,17]. For this reason, digital marketing may encourage deeper and more sustained engagement with BMS and FBHFSS, with potentially negative dietary and health effects [16]. As a result, it is important to identify experiences in systems implemented in countries around the world to monitor the digital marketing of BMS and FBHFSS aimed at ICA.

The present study undertakes a scoping review in order to systematically identify and map the types of methodologies available for monitoring BMS and FBHFSS digital marketing for ICA worldwide. Additionally, it aims to identify where and how these have been conducted and who did so, along with the knowledge gaps in the field. The results will provide an overview of the methodologies employed for monitoring BMS and FBHFSS, provide useful insights for strengthening monitoring, and potentially highlight the need for new modalities to monitor and enforce the legislation regulating the digital marketing of these products.

## 2. Materials and Methods

The format of a scoping review was chosen as the most appropriate method, since it is a type of systematic review that allows for the exploration of the breadth or extent of the literature, map and summarize the evidence, and inform future research [18]. A scoping review maps the available evidence in a systematic way and provides a synthesis of knowledge, including research gaps, that can help in planning future research [19]. The nine-step process of the Joanna Briggs Institute Collaboration (2020) [20] forms the framework for this work, and it was reported according to the guidance provided in the Preferred Reporting Items for Systematic Reviews and Meta-analyses (PRISMA) Extension for Scoping Reviews (PRISMA-ScR) [21].

The concepts of interest were the methodologies and experiences in monitoring the digital marketing of BMS and FBHFSS aimed at ICA. We will use the definition of the WHA Resolution 69.9: ‘marketing’ means product promotion, distribution, selling, advertising, product public relations, and information services [22]. Likewise, ‘digital marketing’ was defined as any promotional activity delivered through a digital medium that seeks to maximize impact through creative and/or analytical methods [23]. Therefore, the following research question was formulated (JBI Approach Step I: defining and aligning the objective and research question): What methods, tools, and techniques have been used to monitor the digital marketing of BMS and FBHFSS for ICA, and which countries have carried them out and how?

The scope review inclusion criteria were defined to clearly establish the basis on which sources will be considered for inclusion (JBI Approach Step II: developing and aligning inclusion criteria with the objective and research question) (Table 1).

To proceed with the search strategy (JBI Approach Step III: describing the planned approach to evidence searching, selection, data extraction, and presentation of the evidence; and Step IV: searching for the evidence), an initial limited search was carried out to identify the keywords from reports and articles related to the topic, which were analyzed to develop index terms regarding BMS and FBHFSS digital marketing monitoring. A second search using identified keywords and combinations was undertaken across all included academic databases (EBSCO, PubMed Central, Web of Science, and ScienceDirect) to identify articles, documents, and reports published 2011–2021 in English and Spanish. The following are the combinations that were used: ‘((monitoring OR digital marketing OR monitoring marketing) AND (unhealthy marketing OR social media OR promotion) AND (digital media OR unhealthy food OR breast-milk substitutes OR children))’. In addition, we considered other non-indexed sources of information—gray literature—. There were no limitations regarding geographical location, country income level, social or cultural group, and journal impact factor. Finally, the reference list of reports and articles, included in the review, was searched for in additional sources.

All titles and abstracts found in the searches were downloaded, and duplicates were removed. The titles and abstracts were independently screened according to the established inclusion criteria by two researchers (V.L.-M. and B.F.-L.). Subsequently, the full-text screening was performed. Any disagreement was discussed by consensus (V.L.-M. and B.F.-L.), if not resolved by the decision of a third reviewer (S.H.-C.). A data charting tool, using Excel spreadsheets, was developed to collect information related to the research question, such as the geographic location, type of methodology, product monitored, platform monitored, and overall results/findings (JBI Approach Step V: selecting the evidence; Step VI: extracting the evidence; and Step VII: analysis of the evidence). Finally, the research team analyzed and mapped the extracted data and discussed the way to present it. The last two steps of the JBI Approach are illustrated in detail in the results section (JBI Approach Step VIII: presentation of the results; and Step IX: summarizing the evidence in relation to the purpose).

## 3. Results

### 3.1. Literature Search

The literature searches provided an initial sample of 753 articles from online databases and gray literature, resulting in 420 unique sources for title and abstract screening. Subsequently, 396 sources were removed for not meeting the inclusion criteria (Table 1), specifically for the use of monitoring techniques aimed at the general population or without a specified population, and the lack of description of the methodology. A total of 24 sources were retained for full-text assessment. After obtaining the full texts, seven sources were excluded: four citations because the target population disagreed with the inclusion criteria; two citations due to incomplete information on the process for monitoring the digital marketing of foods and beverages targeted to ICA; and one citation due to the lack of a full text. Subsequently, 11 additional sources were identified through backward citation chaining and included for data extraction. Finally, 28 sources were considered in this scoping review (Figure 1).

### 3.2. Study Design and Year of Publication

There is growing interest and investigation in the field of the digital marketing of FBHFSS among the under-19 population and BMS for parents with infants. Based on the period studied (2011–2021), literature publications were more frequent from 2016 onwards. After this time, at least one source was published consecutively each year. The highest number of sources was identified in 2020, with six sources within the year (Figure 2) [24,25,26,27,28,29]. Among the sources obtained, peer-reviewed papers (*n* = 20) [25,27,28,29,30,31,32,33,34,35,36,37,38,39,40,41,42,43,44,45] were the most frequent type, followed by official reports (*n* = 4) [46,47,48,49], narrative review (*n* = 1) [50], comments (*n* = 1) [24], open access theses (*n* = 1) [26] and conference abstracts (*n* = 1) [51]. All of the sources included in this scoping review were published in English.

### 3.3. Study Setting

Most of the sources included in this scoping review focused on a single country (*n* = 23); the rest of the publications considered more than one country, with one of them including up to 53 countries [49]. Australia was the country with the highest number of studies (*n* = 5) [31,35,41,45,50], followed by Canada (*n* = 4) [24,30,40,42], the United Kingdom (UK) (*n* = 3) [33,38,49], the Philippines (*n* = 3) [30,46,48], New Zealand (*n* = 3) [36,43,44], the United States of America (US) (*n* = 3) [25,30,34], China (*n* = 3) [26,30,46], Indonesia (*n* = 3) [30,32,46], Thailand (*n* = 3) [32,39,46], Mexico (*n* = 2) [27,31], India (*n* = 2) [40,46], Ireland (*n* = 2) [47,49], Myanmar (*n* = 2) [40,42], Vietnam (*n* = 2) [40,42], and some other countries with only one source (Figure 3). Based on this scoping review, a total of 81 countries have any published documents or reports related to methodologies for the monitoring of the digital marketing of BMS to parents or caregivers (*n* = 27) and FBHFSS to children and adolescents (*n* = 61). Sources for the monitoring of both types of products (BMS and FBHFSS) were available in seven countries (Australia, Canada, UK, US, Philippines, Mexico, and Thailand).

### 3.4. Study Population

Most sources (*n* = 17) focused exclusively on the population of children and adolescents; however, the age range used to define the population differed or was not specified. Among the sources that did specify the age of the population groups (*n* = 7), definitions of age ranges varied between: children aged 7 to 11 years, children aged 13 to 17 years, adolescents aged 16 to 18 years, and young adults aged 14 to 24 years. In those sources where age was not specified, the authors used the terms ‘youth’, ‘children’, and ‘adolescents’. The target population of the remaining included sources (*n* = 11) were parents and families with infants in order to assess their exposure to BMS digital marketing.

### 3.5. Types of Methodologies for Monitoring Digital Marketing

With the purpose of meeting the objective, each methodology for monitoring the digital marketing of BMS and FBHFSS targeted for ICA was identified and categorized as follows:

#### 3.5.1. Visits to Websites or Social Media Sites

There were seven studies that visited the websites and/or social media platforms to conduct BMS and FBHFSS digital marketing monitoring. The first study conducted in the US examined current infant formula (IF) marketing practices on the most visited social media outlets by new and future parents [44]. Firstly, researchers made a list of IF brands widely available in the US, including special medical formulas, as they also fall within the scope of the Code. Then, the most popular social media (Facebook, MySpace, Google, Twitter, YouTube, and mobile applications) were identified, followed by typing the brands’ and manufacturers’ names as keywords in the internal search bar of each. This process was conducted over 6 months to identify IF promotions and related content, including sponsored reviews and mobile applications that were created or enabled by manufacturers and distributors [44]. Each brand’s website was also visited to determine the presence of interactive social media tools (e.g., message boards, photo galleries, and ‘Tell a friend’ tools). Finally, a content analysis was conducted to evaluate promotional practices, using the Code as a basis for ethical marketing. Although the Code applies to a variety of infant feeding products (such as complementary food) and accessories, this analysis was limited to commercial IF [44].

In the UK, the websites of IF manufacturers have also been evaluated. From February to March of 2009, and again in March of 2012, the websites of the top five UK IF manufacturers were visited to identify what kind of online information was provided on infant and follow-on formulas [43]. It should be noted that this was not a formal content analysis of the sites; however, two of the authors independently checked these sites to ensure the reliability of the coding. The mentioned authors had the skills to identify whether there was digital content that would be classified as promotional in nature (such as images of product packaging); whether it was openly stated that breastfeeding is best; and whether it was made explicit that such products should only be used under professional advice (specifically in the case of IF) [43]. With this analysis, it is possible to examine whether information on IF, which cannot legally be advertised to the general public, was as accessible as that on follow-on formulas, which can be advertised openly [43]. Similarly, a Brazilian cross-sectional study also analyzed formula manufacturers’ websites as well as drugstore networks to verify the compliance with Law No. 11,265/2006 (Law for Marketing of Foods for Infants and Toddlers, Feeding Bottles, Teats, and Pacifiers) [29,52]. The selection of the nine drugstore networks was based on the survey of the largest companies in the country conducted by the Brazilian Retail and Consumption Society (2017) and on having an online store. After this selection, the term ‘infant formula’ was entered into the search box, and the predefined items that appeared were examined [29]. Moreover, the five IF manufacturers were chosen based on their presence on all the websites of the selected drugstore networks. The promotion of IFs was examined for a one-month period; the main attributes evaluated were: the use of drawings or representations of children and the presence of pop-up windows with other IFs or links to children’s product websites, among others. Finally, data were obtained on the total number of commercial IFs available on manufacturers’ and drugstores’ websites, as well as on inadequate/non-compliant attributes on the websites [29].

A further study conducted in five countries (Cambodia, Indonesia, Myanmar, Thailand, and Vietnam) aimed to review the regulations and undertake a media audit on the marketing of products within the scope of the Code in South Asia [42]. Independent media agencies were hired to conduct a systematic media monitoring. The media explorations lasted three months in Vietnam and six months in the rest of the countries. The media monitoring included advertisements in print, online, or on television. The keywords used in the research were: breastfeeding, IF, follow-on or toddler milk, growing-up milk, bottles and teats, and milk for pregnant and breastfeeding women, as well as the brand names of popular products. Companies and brands identified through traditional media monitoring in Cambodia, Myanmar, and Vietnam were chosen, and the last 30 posts from their Facebook pages were collected by the researchers. Texts, images and audiovisuals were examined to identify key messages, the stakeholders involved, and the products promoted [42]. To estimate the trend of the market size of the IF industry (from 2000 to 2014), data about the annual growth in Indonesia, Thailand, and Vietnam were purchased from Euromonitor International (http://www.euromonitor.com). Similar data for Cambodia and Myanmar were not available. Additionally, the total annual estimate of ads was calculated by identifying the number of ads and multiplying it by four in Vietnam and by two for the other countries (assuming that ads are distributed equally per month in each country) [42]. Besides being able to assess the compliance of governments and BMS manufacturers and distributors, this methodology can also ensure the comparability of the findings from different countries with the total annual estimate of advertisements. Likewise, it allows for a calculation of the association between the market size, the growth of IF, and the number of advertisements [42].

In this section, the last source found that analyzed the promotion of BMS was a conference abstract of a qualitative study conducted in Europe [51]. The primary objective of such study was to examine the presence of IF marketing on social media. The research consisted of identifying the marketing activities of nine IF brands on social media (e.g., Facebook, Twitter, Instagram), blogs, websites, and mobile applications during the period from December 2016 to January 2018 [51]. Using the WHO Code as a framework and basis for ethical marketing, it was found that companies carry out inappropriate marketing (mainly related to health or nutrition claims) [9,51].

In the case of FBHFSS, a study conducted in the US followed a three-step process to identify the official social media accounts of 200 fast food, beverage, and snack brands with the highest advertising expenditures in the US [25]. This was carried out in order to examine marketing techniques in the top five most popular social media (Instagram, Facebook, Twitter, Tumblr, and Vine) among adolescents and to quantify the increase in social media account creation from 2007 to 2016. First, the Google browser was used to find the home page of the social media. Then, the brand name was entered in the search field available in each social media platform. Finally, the search results were read and the official accounts associated with the brand were searched, identifying the verification badge (e.g., blue or gray checkmark) [25]. Only accounts targeted to consumers in the US were included (e.g., @Coca-ColaUSA). Once the official account was identified, the social media platform name, the number of followers, the number of posts, and the date of their first social media post were collected. In addition, a qualitative analysis was conducted (for a compilation of random posts) through a codebook based on the guidelines described by Lombard and colleagues [53]. By using this methodology with different analysis techniques, it was possible to determine the associations between the use of marketing themes (e.g., holidays) and the presence of interactive tools (e.g., hashtags and requests to comment) in social media posts targeting adolescents. Additionally, it was possible to determine the change in advertising over time (e.g., by adding up the total number of followers considering all the social media of a specific brand) and the nutritional quality of advertised FBHFSS through the developed Nutritional Profile Model (NPM) score [25].

Similarly, an exploratory study was conducted in the UK with the primary aim of quantifying the prevalence of FBHFSS cues (visual and/or verbal display of a food or beverage product/brand) featured in the YouTube videos of the most popular influencers among children (5–15 years) [32]. A secondary objective was to determine the proportion of ‘healthy’ and ‘less healthy’ cues appearing in those videos, in terms of the UK NPM [54]. YouTube videos uploaded by two well-known influencers were assessed using content analysis methods adapted from similar studies [55,56,57,58]. Videos uploaded over a full 12-month period were analyzed to ensure that a representative sample of products was captured, as the content may vary seasonally [32]. The two influencers were neither known for their food expertise nor for blogging about food, but both had previously been involved in FBHFSS digital marketing campaigns on social media. The nutritional information of the products found in the videos was obtained by consulting the companies’ websites or Tesco’s website (the largest supermarket chain in the UK) [32]. Following this process, types of products that were frequently promoted were identified in order to perform a nutritional analysis. Additionally, a content analysis can be conducted to identify important FBHFSS cue characteristics, such as: brand (e.g., supermarket-own brand, unbranded); context (e.g., eating out meal, supermarket, home); description (e.g., positive, negative, or neutral); presentation (e.g., consumed and verbally referenced), and reason for being featured (e.g., paid endorsement) [32].

#### 3.5.2. Screenshots of Posts (Evernote, Full Page Screen Capture)

For data collection, in three of the sources included in this scope review, researchers used posts’ screenshots as the main technique. On one hand, Vandevijvere et al. used Evernote to capture screenshots of every post on 45 company Facebook pages over two months and 15 YouTube channels over two years [37]. Through Socialbakers, the most popular FBHFSS brands on Facebook and YouTube were identified based on the number of ‘likes’ and channel subscribers, respectively. In addition, to estimate the reach of the identified posts on Facebook, the ‘Create Adverts’ feature was used by specifying ‘New Zealand’ and ‘13–18 years’ in ‘Create a Custom Audience’ [37]. Then, the most popular FBHFSS brands, according to Socialbakers, were entered as interests to identify which of them generated the greatest potential reach among participants. Estimates regarding the reach of identified videos on YouTube were not obtained. The followed steps allowed for a determination of the extent, nature, and potential impact of FBHFSS digital marketing among adolescents through the identification of the online behavior (e.g., number of ‘likes’, ‘shares’, ‘comments’ and ‘views’) and marketing techniques (e.g., premium offers, voting, tagging friends, recipe ideas, and the use of cartoons or famous sports person/teams) [37].

On the other hand, a Yale University theses analyzed BMS retail websites on the largest Chinese e-commerce platform (TMall) to characterize the themes and marketing strategies used to target consumers [26]. After identifying the top five domestic and top five foreign BMS company/brand websites of TMall, the main landing page of each flagship website and the product description page of all unique individual formula products (IF, follow-on formula, and toddler milk) were captured. Coding protocols were developed and adapted based on the previous content analysis on the digital marketing of children’s food [59,60]. The 10 main landing pages were coded and separated into two sections (primary and secondary displays) based on the location of the information on the webpage [26]. This study incorporated a combination of quantitative (describing the frequency of different outcomes: thematic appeals, Code violations, and images used) and qualitative (the extraction of specific texts and language to provide context for each variable and for the appeals and marketing strategies targeted to consumers) research techniques in order to improve the understanding of how BMS digital marketing violates Code recommendations and attempts to normalize and promote IF as an alternative to breastfeeding [26].

Similarly, in Australia, websites advertising IF products have been assessed to determine the presence of health and nutritional claims [41]. In order to identify the websites, a one-day search (14 July 2014) was conducted using the Google search engine to approximate consumer behavior. To avoid obtaining results influenced by the researcher’s previous behavior, the browser cache was cleared, and the researcher did not log in before performing the search. The search was limited to Australian sites using ‘infant’ and ‘formula’ as search terms, and then the first 10 pages related with BMS for children aged less than 12 months were selected and examined. All of the pages required visitors to indicate their agreement to directly view pages advertising IF products. The webpages that advertised an IF product or a brand associated with an IF were captured in printed format (using the ‘print this page’ function) or in a screenshot. A thematic coding frame based on the Australian and New Zealand Food Standards Code (F2016C00161) was used to identify nutrient content claims, health claims, and references to the nutrient content of human milk [61]. Lastly, with this methodology, it is possible to identify whether or not websites advertising IF products make a prohibited claim: a health claim (e.g., “reduces abdominal discomfort”), a nutritional content claim (e.g., “contains *l-rueteri*, a beneficial probiotic bacterium”), or a reference to breastmilk (e.g., “high-quality protein found naturally in breastmilk”) [41].

#### 3.5.3. Monitoring through Recording Posts and/or Videos (Lollipop Screen Recorder, iOS Operating System, Tobii Pro Glasses)

A study in Mexico examined the advertising and marketing of BMS through the internet, social media, and television by IF companies over a two-month period [27]. The monitoring was conducted through visits to social media (Facebook, YouTube, and Twitter) or websites of the BMS companies operating on the Mexican market, which were previously identified following the steps that a typical user follows to find them. First, they searched for brand names using the Google browser (e.g., Nestlé in www.google.com.mx), and, subsequently, they conducted a web search using the following keywords: ‘infant formula’, ‘infant milk’, and ‘milk for babies’, among others (in Spanish). Then, the companies’ social media channels (Facebook, Twitter, and YouTube) were found directly on their websites [27]. Only social media sites in Spanish were selected for monitoring. To collect the data, the main home pages were captured using screenshots, and YouTube, Facebook, and Twitter content was recorded. Hashtag frequency on Twitter was measured using the free tracking tool ‘followthehashtag’. Broadly speaking, this methodology allows for the identification of the frequency, scope, and type of BMS promoted in social media in order to assess compliance with local (Mexican laws) or international (the Code) regulations. For example, whether there were texts or images idealizing the use of BMS or discouraging the use of breastfeeding, whether the superiority of breastfeeding was stated, and whether bottles were promoted for using the product, among others [27].

Moreover, the screen recording method was used in a cross-sectional study conducted in Australia to quantify and describe the types of FBHFSS being promoted and the platforms from which the exposures were derived [45]. The screen recording was performed by the participants (adolescents aged 13–17 years old) using their mobile device for two weekdays and one weekend day each time they visited relevant web-based platforms or social media. After each day, they had to upload the video files to a secure server. The recording process varied depending on the operating system of the mobile device, those with Androids downloaded an application called Lollipop screen recorder. For the iOS operating system, the screen recording settings had to be changed from the control panel of the user’s device [45]. The coding frame used in this study allowed for the capturing of both the frequency and duration (seconds) of on-screen promotions, the nature of these promotions (e.g., if it was identified in a paid advertising space or on the food companies’ own websites or transmitted through social media), and any participant engagement. Likewise, considering the WHO European Region NPM guidelines, it enables the designation of products as not allowed or allowed for advertising based on the energy content and the presence of negative nutrients [45].

Similarly, in Ottawa (Canada), an observational study was conducted with children (aged 7–11 years old) and adolescents (aged 12–16 years old) to compare the frequency and healthiness of FBHFSS digital marketing by mobile screen recording [34]. However, in such study, the recording was through Tobii Pro Glasses *2*. The glasses recorded everything in the participants’ field of view during the time they were using two of their favorite social media (e.g., Facebook, Instagram, Snapchat, Twitter, and YouTube) for 5 min each on their cell phone or tablet. Subsequently, the recordings were reviewed to identify and examine different exposures to FBHFSS digital marketing. The categorization of the content analysis makes it possible to identify: food advertisements (e.g., display or video ads); user-generated content (e.g., content produced by a user that intentionally or unintentionally promoted a food brand or product); influencer-generated content (e.g., celebrities that have a following of 10,000 or more), and food marketing embedded in other web content (e.g., branded food products appear in web content such as recipe videos) [34]. Similarly, the healthfulness of the promoted food can be evaluated by the NPM of the Pan American Health Organization (PAHO) and the UK NPM [54,62].

#### 3.5.4. Use of Social Media Analytics Companies or Applications to Select the Most Popular Brand’s Pages (Socialbakers, Create Adverts, comScore, AC Nielsen)

The use of a social media analytics company for sample selection was identified in seven studies. Four studies used Socialbakers, a global social media analytics company, in order to identify, in digital platforms, the most popular FBHFSS brands in a certain period of time. First, a study conducted in Australia used the Socialbakers platform to rank the top 250 Facebook pages with the highest number of ‘likes’ [39]. The aim of such study was to assess the amount, reach, and nature of FBHFSS marketing on Facebook. All non-food and beverage-related pages and those of alcoholic brands were removed. After collecting the number of ‘likes’ and name pages, all posts made by the page since its launch date were identified to carry out content analysis [39]. Using the coding tool, based on previous content analyses of FBHFSS television marketing [59,60], the following marketing techniques can be identified: the use of celebrities, vouchers, games, competitions, apps, user-generated content, and price promotions, among others. In addition, page statistics can be collected, such as: launch date, most popular age group that liked the page, number of people talking about the page, and if consumers could post directly on the page’s wall. The second study was also conducted in Australia with the aim of understanding how sugar-sweetened beverages (SSB) are marketed to young Facebook users (14–24 years) through a content analysis tool (adapted from Carah [63]) of posts made by popular brands [35]. The research team used the Socialbakers site to rank the top 20 Australian-specific SSB pages based on the total number of ‘likes’. Global and brand-specific pages for artificially sweetened beverages were excluded. Finally, all official posts made by the six most popular pages in each category (soda, sports drinks, and energy drinks) over a six-month period were collected [35]. This methodology was used to identify ‘call to action’ strategies (techniques where brands encourage their followers to do something, such as ‘likes’, ‘comments’ and ‘shares’), ‘hashtags’, and thematic analyses of posts [35].

In a similar way, the third study was conducted in Thailand with the objective of quantifying the magnitude and profile of FBHFSS digital marketing content on Facebook directed toward children and youth [33]. Using the Socialbakers platform, the 30 most popular confectionery, soft drinks, and retail food Facebook pages were selected. The pages were recorded for 24 h over the course of a month and saved as PDF files for content analysis. To analyze the data collected, the authors adapted a previously validated tool for the evaluation of FBHFSS in television marketing [13]. This methodology seeks to identify marketing techniques and tactics (usage of pictures, branding elements, ‘hashtags’, conversations, special promotions, links, videos, competitions, prizes, giveaways, branded characters, celebrities, games, apps, and several others) on Facebook to see if the marketing content complies with government regulations and industry self-regulatory codes [33]. The fourth study analyzed the nature of Mexican FBHFSS/brand digital media with the greatest number of followers on social media (Facebook, Instagram, and Twitter) and websites with specific appeal to children and adolescents [31]. The process included, as a first step, the identification of the companies’ profiles with the largest audience on social media (Facebook, Instagram, and Twitter) using the Socialbakers platform. The following step consisted of selecting, for each social media app, the top 10 products/brands from three groups: ‘Soft drinks’; ‘Fast Moving Consumer Food’ (e.g., chips, yogurt); and ‘retail food’. The third step consisted of excluding duplicates and integrating a unique list. The fourth step identified the social network accounts of the selected products. Finally, the fifth step consisted of excluding accounts with no activity in the month prior to data collection. Following this five-step process and utilizing the coding system, which was previously developed by Mexican and international teams [60,64,65], the following information can be identified: product/brand information; media information; associations with other brands/products; featuring of characters (e.g., cartoons, famous athletes, influencers); incentives (e.g., gifts, samples, concert tickets); downloads of apps or games related to the promoted product; and digital techniques (advergames, hashtags). This had the aim of identifying persuasive techniques and the nutritional quality of the FBHFSS promoted—in this case, according to the PAHO NPM criteria [31,62].

Since Socialbakers was no longer free in 2015, other social media analytics companies, such as AC Nielsen and comScore, have been used in the literature. Tatlow-Golden et al. used the AC Nielsen ranking and Facebook’s Create Adverts feature [47]. The first part of the study involved locating retail FBHFSS brands that appeal most to young teens (aged 13–14 years old) in Ireland using the AC Nielsen ranking. Subsequently, Google searches were conducted to identify brand websites, and, then, over 13 days, a full ‘sweep’ was performed to identify the content that appealed to young people and parents (e.g., teen activities, the use of sporting celebrities and competitions with entertainment-, media- and sport-based prizes) [47]. Then, the second part examined FBHFSS brands’ Facebook pages with the highest ‘reach’ among young teens through Facebook’s Create Adverts. This feature defines a target audience for potential advertisers and can be selected from the dropdown menu when logged in to Facebook as the Page Administrator of a youth-oriented page. In this case, the researchers in ‘Create a Custom Audience’ specified the country and the population age. The ‘Interests filter’ then lists the size of specific audiences by looking at their interests and online behavior (e.g., Facebook evaluates the potential audience that has ‘liked’ the named pages). With the objective of identifying which of the FBHFSS brands have a greater reach among young teens, the brands most ‘liked’ by the general population were entered as ‘Interests’ [47]. Finally, the 20 previous posts of each selected brand were collected and analyzed, no matter how far back in time. In addition, page ‘likes’ and shared posts were analyzed. In brief, following these steps, which include the Create Adverts feature, the relative popularity of specific brands on Facebook can be measured to perform a content analysis of FBHFSS posts targeted at young people [47].

Moreover, a study conducted in New Zealand evaluated the most popular websites among children (aged 6–12 years old) and adolescents (aged 13–17 years old) by purchasing internet traffic data from the market research company AC Nielsen [38]. The aim of such study was to measure the extent and nature of FBHFSS promotion through the internet. The websites with an audience greater than 1.5% (>10,365 children and adolescents) of the target population were included. However, the list included mainly non-food websites, so additional FBHFSS brand websites most frequently marketed through television, sports, magazines, and Facebook were selected (derived from previous New Zealand studies and Socialbakers) [66,67,68]. Marketing techniques and related features used on food brand websites were recorded for a period of 2 months for further analysis [38]. Two coding tools were adapted from a previous Australian study, one for non-food websites and one for food brand websites [60]. All FBHFSS products marketed on websites were classified according to the New Zealand Ministry of Health Food and Beverage Classification System. With this methodology, a range of marketing techniques can be identified, such as promotional characters; nutrition and/or health claims; and one of the newest techniques such as ‘advergaming’, by analyzing the games on food brand websites to identify the presence of food products, brand logos, spokes characters, music, sound effects, and animation during the games [38].

A Canada-wide study (with the exception of Quebec) used comScore’s Ad Metrix module to estimate internet advertising exposure data and the behavior of the internet-using population [36]. comScore is a company that has information from approximately 40,000 Canadians about the websites they visit and the advertisements that appear on those sites. After a period of four months of using comScore’s Ad Metrix Key Measures Report, the 10 most popular websites with advertising targeting adolescents (12–17 years old), including ads from out-of-country-based food companies, were determined [36]. The most popular websites were defined as those with a minimum of 50,000 adolescent visitors. Subsequently, by generating comScore’s Ad Metrix Advertiser Report, the frequency of food ads on each selected website was identified. The nutritional analysis of the products found was based on the NPMs of PAHO and the UK [54,62]. In general, by using social media analytics companies (comScore’s Ad Metrix module, Socialbakers, and AC Nielsen) or a feature within app (Create Adverts) as a first step in the papers included in this section, it was possible to identify the most popular FBHFSS brands and, based on content analysis, determine the marketing techniques used and the impact (measured by ‘likes’ and ‘shares’). In addition, it was possible to classify the healthiness or unhealthiness of the products advertised [36].

#### 3.5.5. Browser Extension to Collect Advertisements (AdHealth)

A cross-sectional study conducted in New Zealand used a browser extension (AdHealth) on Facebook with the aim of testing the feasibility of a methodology to estimate the exposure of adolescents (aged 16–18 years old) to FBHFSS advertisements on Facebook [30]. AdHealth was developed in JavaScript by the University of Auckland’s Centre for eResearch and was made available to participants via a promotional website marketed on Facebook [69]. Upon approval of their participation, they downloaded the extension and it automatically linked to their Facebook account on their personal computer, either desktop or laptop computer, as the extension did not work with a smartphone or tablet. The extension could be removed at any time, so participants could withdraw from the study. Over a period of 1535 days, advertisements seen by adolescents when scrolling through their newsfeed were recorded in an online database [30]. Variables of interest collected by AdHealth for each advertisement entry included: user ID, session ID, advertisement ID, and URL to view the advertisement. This methodology allowed for a characterization of the types of persuasive techniques (e.g., display of cartoons/company-owned characters, premium offers as a price discount, among others) used in food-related advertisements on Facebook and the healthiness or unhealthiness of the advertised food according to the WHO-EURO NPM [70].

#### 3.5.6. Web-Based Structured Questionnaire

A study conducted in the Philippines examined the marketing tactics of BMS companies since the onset of the COVID-19 pandemic through a quantitative methodology by obtaining data from an official database of reported violations of Executive Order 51 (the Milk code) [40]. Database reports, regarding violators, product types, types of violations, and channels were obtained using mobile and web-based platforms with standardized form fields. All of the population groups were able to respond to the online questionnaire and report advertisements contravening the Code, which were identified either in establishments at health facilities or mass media. In addition, this study included a qualitative arm, in which 26 examples of 9 companies from 14 countries (Burkina Faso, Canada, China, India, Indonesia, Kenya, Laos, Malaysia, Myanmar, Pakistan, Singapore, Philippines, United States, and Vietnam) derived from the internet (e.g., infant feeding and child nutrition blogs, social media, company websites) and print magazines were analyzed [40]. The collection period was from August to October 2020, just after the WHO declared COVID-19 a “public health emergency of international concern”. Among all the data collected, only examples that directly or indirectly referred to the COVID-19 pandemic were selected. The study used thematic analysis, which allowed them to identify and describe categories and themes. Additionally, a constant comparison technique was used to inductively compare emerging categories [40].

Furthermore, a cross-sectional study was conducted in two urban areas in the Philippines (Baguio and Cebu) to identify the marketing strategies of FBHFSS on social media (Facebook, Instagram, and YouTube) targeting children and adolescents (aged 5–17 years) [48]. The research team recruited participants through public Facebook and Instagram posts directed to parents and guardians, providing information about the purpose of the study. Due to the COVID-19 pandemic, the study was conducted online via Zoom, with an open-ended and closed-ended questionnaire focused on internet device usage, social media choices, and awareness of advertisements. Additionally, the study conducted a second arm to analyze FBHFSS products and brands on social media. The research team selected the most used social media in the country (Facebook, YouTube, and Instagram) and identified a set of FBHFSS brands popular among Filipino consumers. To identify brands or key products relevant for under-18 audiences, a Facebook Ad Manager analysis was carried out, and the sales of the product were considered. A total of 20 different posts or videos of each brand were analyzed to determine the healthiness of the product. Nutrient analysis was performed based on the WHO NPM for the Western Pacific Region (2016) [70]. With this methodology, it is possible to identify the social media preferences of children and, in turn, assess whether the ads they see on them should be banned from advertising [48].

#### 3.5.7. Digital and Social Media Ethnography Approach

In South Africa, a study used a digital and social media ethnography approach with the aim of providing examples of how BMS manufacturers use social media (Facebook and Instagram) to market their products and violate national regulations [28,71]. The purpose of this is to study, and the ethnography approach was intended to explore BMS organizations’ activity on social media platforms. First, one of the authors identified sponsored posts on their own Facebook newsfeed and then actively searched for different Facebook pages and Instagram accounts managed by BMS manufacturers in the country (from 2015 to 2019). Previously, the author had already ‘liked’ or ‘followed’ pages related to IYCN. Examples of social media posts that the author has observed were included, as they most commonly appear as sponsored posts and are clear examples of violations of National Regulation R991 (because designated products are those targeted to children under the age of 36 months). Therefore, the selected posts were interpreted according to national provisions, and an image and/or web link was provided to illustrate the violation. Following the aforementioned methodology, it is possible to purposely select examples of publications observed on social media and perform a content analysis in terms of national legislation [28].

#### 3.5.8. ‘WHO CLICK’ Monitoring Tool

A report based on the meeting on monitoring the digital marketing of FBHFSS to children and adolescents, organized in 2018 by the WHO European Office for the Prevention and Control of Noncommunicable Diseases, was included. Based on that meeting, in 2019, a tool was developed to support Member States of the WHO European Region in monitoring the digital marketing of FBHFSS to ICA [49]. The CLICK tool is a five-step process based on the following acronym: ‘C’ (Comprehend the digital ecosystem); ‘L’ (Landscape of campaigns); ‘I’ (Investigate exposure); ‘C’ (Capture on-screen), and ‘K’ (Knowledge sharing). This combination of different methods can provide a much deeper insight into and richer explanation about what is currently available. This monitoring framework is flexible and can be adapted to the national context [49].

The first step is to map the in-country stakeholders and marketing ecosystem to obtain an initial understanding of children’s online website and app habits (‘C’) [49]. The next step is to assess the ad campaigns of products/brands of the type of advertising to which ICA are exposed (‘L’). Subsequently, ICA’s interaction with advertisements on some websites and social media was mapped by a panel of children (of different ages) through an application installed on a smartphone (‘I’). Then, the next step is to use real-time screen capture software to assess what a representative sample of ICA see online on their devices in order to better understand marketing techniques (‘C’) [49]. Finally, the last step is to create user-friendly materials based on the research data and develop partnerships (including ICA, parents, policy makers, and civil societies) to raise awareness and influence policy (‘K’). This methodology can collect information on the amount of time spent on each platform, the number of ads clicked, the brands advertised, activities with the ad (e.g., whether the video advertisement is played, paused, or skipped after five seconds), if an ad is paid or user-generated, etc. [49].

#### 3.5.9. IBFAN-ICDC Code Monitoring Toolkit

A report with a compilation of the Code monitoring findings from 11 Asian countries (Bhutan, China, India, Indonesia, Republic of Korea, Maldives, Mongolia, Nepal, Philippines, Sri Lanka, and Thailand) was found [46]. The International Baby Food Action Network (IBFAN) and International Code Documentation Center (ICDC) provided technical support in each country monitoring exercise through the development of online forms and databases, the compilation of reports, and with the participants’ training process required. In order to identify the presence of the Code violations (advertisements, discounts, free samples, and free gifts, among others) and their predominant trends, they created a monitoring methodology called the ‘Code Monitoring Toolkit’ for sharing the findings of each country with IBFAN-Asia and IBFAN-ICDC [46]. This tool shows the steps needed to carry out monitoring on the labels of the products covered by the Code and on the e-marketing portals. To monitor Code violations on e-marketing websites, the following process was carry out: (1) four popular e-marketing websites in the respective countries were examined (e.g., Amazon and eBay, among others) to identify advertisements and other marketing techniques (e.g., providing the company’s contact details on the portal and asking mothers to get in touch in case they had any questions or sought help), and (2) the information thus generated was sent to the ICDC using a form previously developed (Online Quick and Easy Code Monitoring Form). Each person working as a monitor in the country received a password to facilitate submission. Attached to the complaint on the designated form, ‘photographs’/’screenshots’ of the reported violation were submitted as supporting material. With the information obtained from the countries, a national and regional report of the Code violations regarding labeling and promotion on e-marketing websites for baby food and baby bottles can be derived [46].

#### 3.5.10. Proposal of New Methodologies (‘WHO CLICK’ + Artificial Intelligence, Step-Wise Framework)

The ‘WHO CLICK’ monitoring framework is very useful, as it explains how a combination of methods can facilitate the monitoring of FBHFSS and brands’ digital marketing targeting children; however, it does not include Artificial Intelligence (AI)-enabled tactics [49]. Olstad and Lee reported this omission as being important, since AI is currently the only viable way to address the massive volume, variety, and dynamism of digital marketing [24]. Additionally, it has been considered essential for research and policy making in the digital domain, as it can enable researchers to efficiently and accurately automate processes, such as: extracting certain characteristics from text, images, and videos [24]. Similarly, AI can facilitate mapping foods to nutritional information, identifying targeted marketing to a certain population, and classifying marketing strategies [24]. Therefore, they proposed a seven-step process in which AI complements the ‘WHO CLICK’ monitoring framework, providing a more complete picture of the frequency and nature of FBHFSS digital marketing to ICA. As well as a solid basis for monitoring compliance with the policy to help ensure that ICA receive maximum protection. The process considers seven steps in total, of which three are computational or rules-based algorithms (Marketing feature extraction, Healthfulness classification, and Policy adherence classification) and two machine learning models (Food and Brand identification and Marketing strategy classification). The above five steps should be repeated for each marketing instance. Finally, the other two steps correspond to bulk operations (Marketing data collection and Aggregation and visualization) [24].

A study conducted in 2013 proposed a ‘step-wise framework’ for monitoring the frequency and level of the exposure of population groups (especially children) to food promotions, the power of promotions, and the nutritional compositions of the promoted products [50]. This framework includes monitoring activities that are part of the ‘minimal’, ‘expanded’ and ‘optimal’ approaches. In general terms, the ‘minimal’ approach involves measuring children’s exposure to media promotions during a limited number of time points and should focus on younger children (under 12 years old). The ‘expanded’ approach attempts to assess the exposure of children and adolescents to promotions across several dominant media over more time points. Finally, the ‘optimal’ approach measures both the extent of exposure and the power of promotions across all dominant media. The results of the stepwise framework should be compared with existing national policies on food marketing to children to assess the degree of implementation and the extent to which they are effective [50].

The key general characteristics of all the studies are presented in Figure 4, as well as some specific recommendations based on the gaps identified in the methodologies. In addition, specific information for each of the studies included in the scoping review (Table 2) is shown below.

## 4. Discussion

This scoping review identified studies monitoring the digital marketing of BMS and FBHFSS targeted to ICA. The studies, mostly published in the last 5 years, applied different approaches and methodologies in about 80 countries, with the vast majority of them focused on studying FBHFSS digital marketing. Most of the included studies aimed at understanding and describing BMS or FBHFSS digital marketing techniques and strategies, as well as identifying the nutritional content, according to the NPM. While studies used several methodologies to identify these products in digital media, the majority have been used primarily for research. There were few studies exploring the feasibility of implementing strategies to monitor exposure to BMS or FBHFSS digital marketing [24,30,49,50]. There is an urgent need for a cost-effective monitoring tool to establish, as far as possible, the reality of pregnant women, mothers of children younger than 36 months, the population targeted by BMS promotion, as well as children’s and adolescents’ exposure to BMS and FBHFSS digital marketing. Additionally, there is a need to identify non-compliance with the provisions of national legislation or international instruments (e.g., the Code) and to hold the party accountable.

The urgent call for a monitoring system for BMS and FBHFSS marketing on digital environments is given by the marketing techniques’ ability to increase consumer demand and consumption, with the corresponding risk for adequate feeding practices (e.g., breastfeeding or intake of healthy foods and beverages) [72,73], health consequences, and economic loss [74,75,76]. This marketing influence on purchasing decisions has also been described for tobacco and alcohol [77,78]. Industry spending on marketing, accounting for 5–10% of annual turnover, also reflects this confidence [79]. In many countries, digital marketing is becoming the dominant form of marketing; for instance, over 80% of advertising exposure for BMS occurs online [23]. It is critical to ensure an overall environment that enables pregnant women, mothers of children under 36 months, children, and adolescents to make the best possible decisions about feeding, based on unbiased information and free of commercial influences, and to be fully supported with an adequate environment in doing so.

Another important finding from this scoping review is that most of the included studies aimed to understand the extent and potential impact of digital marketing and the user engagement with online content (e.g., call-to-action strategies such as ‘liking’ or ‘sharing’ their posts, tagging, participating in events, among others) and display advertisements (e.g., pop-up banner ads). For pursuing the former, most of the studies carried out content analysis. Some other studies explored digital marketing techniques and strategies according to local regulations or international frameworks (e.g., the Code for studies addressing BMS promotions), while another important proportion of studies included in this scoping review aimed to characterize NPM, which can support adequate regulatory practice as part of obesity and chronic disease prevention policies [80]. The research of these issues is crucial in order to understand the context of the digital marketing ecosystem, as suggested by the CLICK methodology proposed by the WHO Regional Office for Europe, and to identify key areas for monitoring [49]. Interestingly enough, even though the use of new technologies is very innovative and an essential feature of a digital marketing monitoring system that can be scaled up and sustainable, only in some studies were applications used to monitor digital marketing [30,34].

One aspect that is important to highlight is that, among all the studies identified in this scoping review, just three of them explored the FBHFSS promotion through influencers [31,32,34]. Coates et al. reported that, of the 380 analyzed YouTube videos, 92.6% of them featured FBHFSS cues. Cakes and fast foods were the most frequently featured products, and less frequent were healthier products such as vegetables and fruits [32]. None of the publications with methodologies used to monitor BMS considered influencer-generated content. The above is noteworthy because a recent BMS digital marketing study in several countries described the use of influencers as a marketing strategy that is becoming increasingly prevalent [23]. In addition, regarding studies addressing BMS digital marketing, only one of them focused on drugstore websites [29], and none focused on online baby clubs, indicating an important monitoring gap given that this is another source of BMS promotion [23].

Several methodologies for monitoring FBHFSS were identified, some with a combination of methods and indicators to achieve a more comprehensive scope, such as the previously described ‘WHO CLICK’ methodology [49]. Similarly, a methodology was developed in 2013 by the International Network for Food and Obesity/NCDs Research, Monitoring and Action Support (INFORMAS) [81]. This works as a framework to monitor and benchmark the healthiness of national food environments and policies in a standardized way, which consists of nine modules with indicators related to nutritional policies and the food environment (e.g., food composition, labeling, marketing, food in public sector settings, food retail, among others). Such indicators are related to the extent and nature of FBHFSS marketing targeted at children through several media (e.g., television, magazines, websites, Facebook, food packages, children’s sport clubs, and around schools). Up to 2019, around 30 countries had implemented one or a combination of those modules to provide a comprehensive picture of different food environments [82]. During our review, New Zealand papers describing the methodology for digital marketing monitoring were found and included in this scoping review [37,38].

It is important to highlight that, in this scoping review, fewer studies (*n* = 11) were found that illustrated research methods for monitoring the digital marketing of BMS. Most of the studies (*n* = 13) focused on monitoring the marketing of FBHFSS targeted toward adolescents. The latter finding is of relevance, considering the influence of BMS marketing on infant feeding practices and the number of WHO Member States that have not yet adopted legal measures to implement at least some of the Code’s provisions [73]. Of the 144 member states, 28.5% (41 countries) are moderately aligned with the Code, and 22.2% (32 countries) are substantially aligned [11]. Recent evidence shows aggressive marketing strategies to reach pregnant women and mothers, the target population for these promotions [11]. For instance, companies are trying to reach younger and newly pregnant women by using data-driven algorithms that target digital advertising to women whose online behavior suggests they may be pregnant [83]. Thus, society, health professionals, and governments need to report the unethical marketing of IF to a much wider audience and take decisive action to end these practices.

To our knowledge, this is the first scoping review addressing and summarizing the methodologies for monitoring the digital marketing of BMS and FBHFSS. Although our search has some limitations, this study provides an overview of the methodologies usually employed for monitoring and analyzing these types of products in the digital environment. In addition, it emphasizes the need for an accessible monitoring system that generates reproducible and comparable results of digital media advertising. Limitations of the study worth discussing are as follows. Firstly, we restricted our search strategy to publications in English and Spanish, giving the possibility of missing information published in other languages. Secondly, our search may not have captured studies not accessible in online databases or from researchers without funding for their publications. However, these were mitigated by our search of multiple databases, the absence of search restrictions on the geographic location, country income level, or journal impact factor, and backward citation chaining. Another limitation is the lack of included studies’ quality assessments, although this is unnecessary for scoping reviews.

Based on our findings on the methodologies used to monitor BMS and FBHFSS, some specific recommendations are the application of AI to efficiently and automate processes, as recommended by Olstad and Lee [24], as well as the inclusion of new social media apps such as Tik Tok and other platforms as online retailers and baby clubs. Further recommendations are given in Figure 4. In addition, stopping unethical BMS and FBHFSS digital marketing needs to be a priority across society, and not just from groups and individuals involved in the nutrition and health of ICA. It is critical to involve civil societies and consider the challenges that may arise—for example, the different promotions and subtle ways of marketing that make it difficult for the population (e.g., pregnant women, mothers of children under 36 months, children, and adolescents) to recognize it. Furthermore, given the complexity of the system, a self-regulatory algorithm to know what ads an individual user looks at on their device is not currently available to any of the organizations involved in this multi-step process (e.g., publisher, media agency, or brand). No one, not even the brands themselves, can assess which people see their ads, where they are, or on which device they see them. Therefore, even if companies enter into voluntary agreements to restrict their advertising to children and adolescents, in the current ecosystem, no brand has the power to fully control the process.

Efforts have been made to involve citizens in identifying inadequate marketing practices, particularly for BMS promotion. This is the case with a monitoring system carried out in Brazil and the Philippines. The ‘National System to Monitor the International Code of Breast Milk Substitutes’, implemented in Brazil, is an internet-based program intended to report Code violations in retail stores and health facilities [84]. The application of this monitoring system allows citizens, governmental agencies, and research institutions with internet access and awareness of the issue to report the violations identified. However, to our knowledge, the latter does not include the reporting of violations related to digital media [84]. Unlike the monitoring system implemented by Brazil, the Philippines also has a monitoring system that allows citizens to report breastfeeding-related law violations observed on the internet and digital media. Currently, this system receives reports not only via internet connection but also through Short Message Service (SMSs) [85]. In addition to monitoring systems, awareness-raising campaigns on the marketing strategies commonly used by the food industry are needed to successfully involve the general public in the monitoring of inappropriate and unethical marketing.

We identify some opportunities for action. First, senior political leaders, public health institutions, health professionals, and civil societies must fully recognize and expose the pervasive and invasive nature of the digital marketing of food and beverages and the harm it causes to ICA. The former has been clearly stated for both BMS and FBHFSS, along with the importance and urgency of regulating the inappropriate marketing of such products [8,86]. It is important to recognize how this unethical marketing is a human rights issue (e.g., the right to adequate food, to have an enabling environment to enhance access, and to receive adequate information to make an informed decision) [87], as well as a threat for societies, economies, and the environment. Second, countries should urgently adopt or strengthen national mechanisms to prevent BMS and FBHFSS digital marketing. Third, the whole digital ecosystem should be subject to a thorough review from a public health perspective. Governments and international authorities must develop enforceable regulations that protect the health and development of ICA from harmful digital marketing. Fourth, the use of globally benchmarked product and service expenditure databases, such as Euromonitor or similar databases, could be considered as a complementary tool in the monitoring of BMS and FBHFSS. Equally important is the use of this monitoring as a tool to achieve accountability in cases of breaches to local legislation. Lastly, civil society should be involved in denouncing the inadequate promotion of BMS and FBHFSS. Public awareness of the strategies used by companies to promote and sell their products should be raised, and spaces where complaints can be lodged should be created.

## 5. Conclusions

Digital marketing represents new challenges for the protection of breastfeeding and healthy eating at an early stage, calling for a review of local laws to prevent inappropriate promotion practices of BMS and FBHFSS. It is equally important to identify strategies and tools for monitoring such practices across the vast number of social media platforms and mobile devices and to consider all the various strategies of digital marketing. Thus, it is paramount to develop a feasible and scalable monitoring system to supervise the extent of inappropriate marketing in order to develop effective policies to protect pregnant women, mothers of children under 36 months of age, children, and adolescents from these unethical BMS and FBHFSS promotions, as well as to ensure policy adherence.

## Figures and Tables

**Figure 1 ijerph-19-08951-f001:**
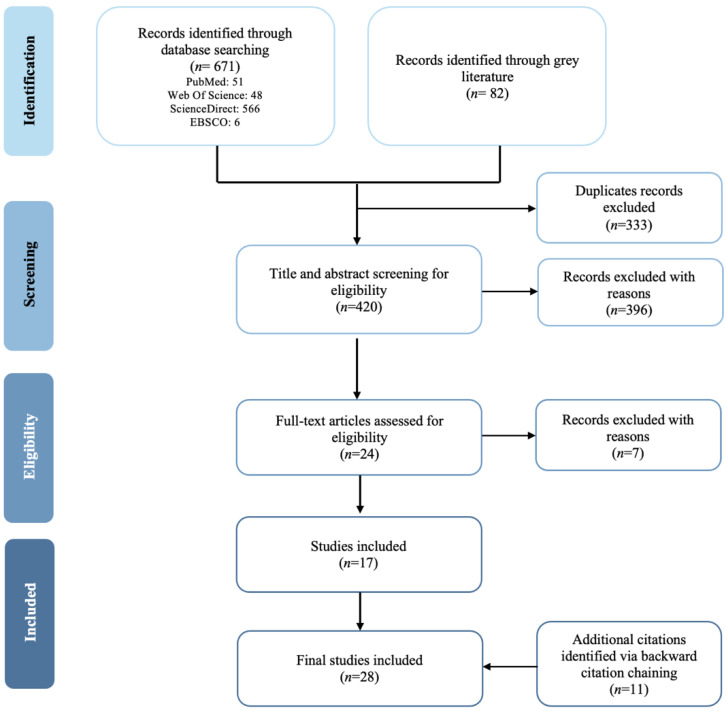
Preferred Reporting Items for Systematic Reviews and Meta-Analyses Extension for Scoping Reviews (PRISMA-ScR) flow diagram illustrating the search strategy.

**Figure 2 ijerph-19-08951-f002:**
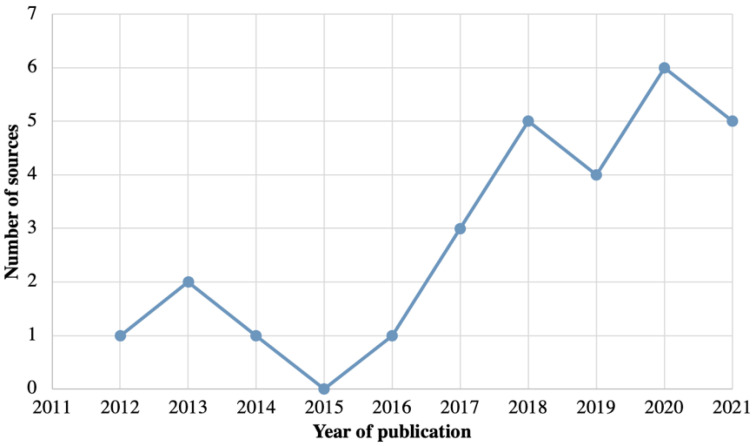
Number of sources published from 2011 to 2021 on the digital marketing monitoring of BMS and FBHFSS targeted for ICA (*n* = 28). BMS: breast-milk substitutes; FBHFSS: foods and beverages high in saturated fat, salt, and/or free sugars; ICA: infants, children, and adolescents.

**Figure 3 ijerph-19-08951-f003:**
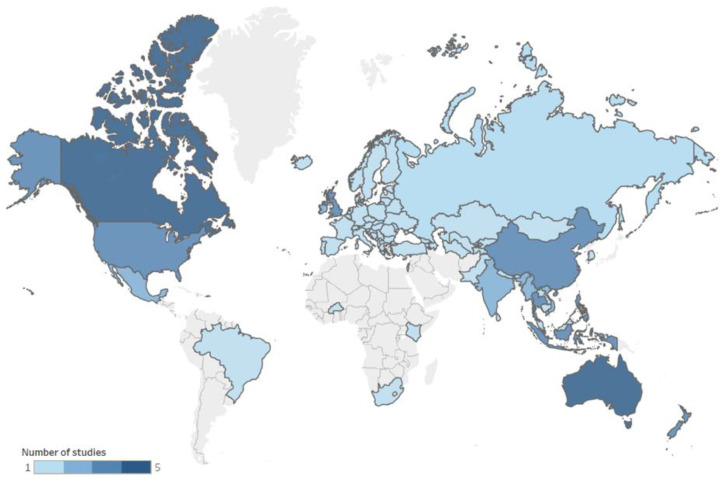
Source countries on BMS and FBHFSS digital marketing monitoring targeted for ICA (*n* = 81). BMS: breast-milk substitutes; FBHFSS: foods and beverages high in saturated fat, salt, and/or free sugars; ICA: infants, children, and adolescents.

**Figure 4 ijerph-19-08951-f004:**
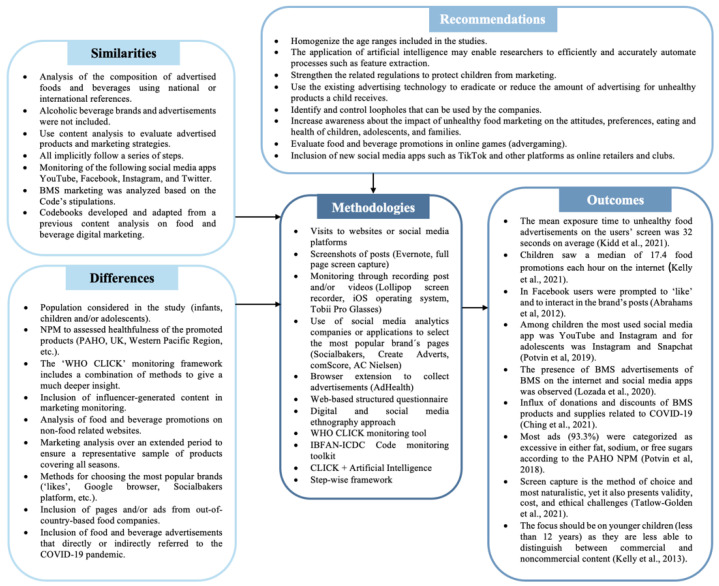
Key characteristics of the included studies on methodologies for monitoring BMS and FBHFSS digital marketing to ICA [27,30,34,36,40,44,45,48,50]. BMS: Breastmilk substitutes; the Code: International Code of Marketing of Breast-milk Substitutes; NPM: Nutritional Profile Model; PAHO: Pan-American Health Organization, UK: United Kingdom; WHO: World Health Organization; BMS: breast-milk substitutes; FBHFSS: foods and beverages high in saturated fat, salt, and/or free sugars; ICA: infants, children, and adolescents; IBFAN: The International Baby Food Action Network; ICDC: International Code Documentation Centre.

**Table 1 ijerph-19-08951-t001:** Scoping review inclusion criteria.

Types of Evidence Source	The Type of Evidence Is ‘Open’ to Allow for the Inclusion of All Types of Sources (e.g., Primary Research Studies, Systematic Reviews, Meta-Analyses, Reports, Letters, Guidelines, Websites, and Blogs, among Others).
Concept	Sources describing methodologies and experiences in monitoring the digital marketing (e.g., promotional activities through websites, online retail platforms, online gaming, online groups, social networks such as Facebook, Instagram, and TikTok, micro-blogging services such as Twitter, and content communities such as YouTube) of BMS and FBHFSS targeted to the under-19 population
Population studied	Infants (ages 0–1 years), children (ages 2–9 years), and adolescents (ages 10–19 years)
Context	Worldwide settings
Timeframe	1 January 2011 to 31 October 2021
Language	English and Spanish
Access	Full-text article accessible

BMS: breast-milk substitutes; FBHFSS: foods and beverages high in saturated fat, salt, and/or free sugars.

**Table 2 ijerph-19-08951-t002:** Summary of the included sources in the scoping review (*n* = 28).

Author (Year)	Region/Country	Media	Advertising Target Population	Objective	Description
Breast-Milk Substitutes (BMS)					
Ching et al. (2021) [40]	Burkina Faso, Canada, China, India, Indonesia, Kenya, Laos, Malaysia, Myanmar, Pakistan, Singapore, Philippines, United States (US), and Vietnam	Facebook, Instagram, and company website	Parents and families with infants	To examine the marketing tactics of breast-milk substitutes (BMS) companies since the start of the COVID-19 pandemic.	Data were collected based on a structured questionnaire and social media posts collected in several countries after the onset of the COVID-19 pandemic. A quantitative analysis of the information from the questionnaire and a thematic analysis of the posts were carried out.
Han (2020) [26]	China	e-commerce platform TMall	Parents and families with infants	To analyze the BMS retail websites on the largest business-to-consumer (B2C) e-commerce platform, TMall, to characterize the marketing themes and strategies used to target consumers.	Use of a Chinese e-commerce platform to collect the top ten BMS company/brand websites. A coding tool was developed based on previous web-based content analysis on the marketing of children’s food, including thematic appeals, International Code of Marketing of Breast-milk Substitutes (‘the Code’) violations, and images used.
Lozada et al. (2020) [27]	Mexico	Facebook, Twitter, and YouTube	Parents and families with infants	To examine the advertising and marketing of BMS through the internet, social media, and television in Mexico.	Recording of posts of the main BMS companies on social media. Content analysis of BMS posts based on the Code’s stipulations.
Pereira-Kotze et al. (2020) [28]	South Africa	Facebook and Instagram	Parents and families with infants	To provide pertinent examples of how BMS manufacturers in South Africa use social media to market their products and how this marketing relates to existing national regulations.	Use of a digital and social media ethnography approach to identify and analyze BMS posts (purposely selected) according to the provisions of Regulation R991 (national legislation).
Prado & Rinaldi (2020) [29]	Brazil	Formula manufacturers’ websites and drugstore networks	Parents and families with infants	To verify the compliance with Law No. 11,265/2006 in the promotion strategies for IF on Brazilian websites of manufacturers and drugstore networks.	Five websites of IF manufacturers and nine websites of drugstores networks were selected to analyze compliance with national regulations. The main attributes evaluated were: the use of drawings or representations of children, the presence of pop-up windows with other IFs or links to children’s product websites, among others.
Senkal & Yildiz (2019) [51]	European country	Facebook, Twitter, Instagram, blogs, websites, and mobile apps	Parents and families with infants	To examine the presence of IF marketing on social media	Identification of the marketing activities of nine IF brands on social media, using the WHO Code as a framework.
IBFAN Asia & IBFAN-ICDC (2018) [46]	Bhutan, China, India, Indonesia, Republic of Korea, Maldives, Mongolia, Nepal, Philippines, Sri Lanka, and Thailand	e-marketing portals (Amazon, eBay, among others)	Parents and families with infants	To identify the presence of Code violations and their predominant trends in labeling and online portals.	Report regarding the Code violations on popular e-marketing portals. Data collection was based on its own toolkit, and a theme-based approach analysis was conducted on the stipulation of the Code.
Berry & Gribble (2017) [41]	Australia	Websites advertising IF products	Parents and families with infants	To determine whether such prohibited claims could be observed on Australian websites that advertise IF products.	Through a one-day search, 25 websites were identified. A thematic coding frame based on the Australian and New Zealand Food Standards Code was used to identify nutrient content claims, health claims, and references to the nutrient content of human milk.
Vinje et al. (2017) [42]	Cambodia, Indonesia, Myanmar, Thailand, and Vietnam	Editorial content, Facebook, and Television	Parents and families with infants	To review regulations and to perform a media audit of the promotion of products under the scope of the Code in South-East Asia.	Media monitoring of advertisements in print, online, or on television by independent media agencies. The last 30 of Facebook posts were identified, and texts, images, and audiovisuals were examined to identify key messages, the stakeholders involved, and promoted products.
Gunter et al. (2013) [43]	United Kingdom (UK)	Formula manufacturers’ websites	Parents and families with infants	To examine formula manufacturers’ web sites to ascertain whether these are used as alternative forms of advertising that fall outside current regulations.	The websites of five major IF manufacturers were evaluated for the presence of text and images related to IF products that are not allowed to be advertised directly to consumers under the current regulations.
Abrahams et al. (2012) [44]	US	Facebook, MySpace, Google, Twitter, YouTube, Mobile Applications	Parents and families with infants	To examine the presence of IF marketing on social media sites that are likely to be visited by new and expectant parents, in order to describe how social media are used for the promotion of breast milk substitutes in the US.	The most popular social media (Facebook, MySpace, Google, Twitter, YouTube, and mobile applications) of BMS companies were identified and examined. A content analysis of the posts was conducted based on the Code’s stipulations.
Foods and beverages high in saturated fat, salt, and/or free sugars (FBHFSS)					
Kelly et al. (2021) [45]	Australia	Instagram, Facebook, Snapchat, YouTube, and Twitter	Adolescents (13–17 years)	To monitor the extent of children’s exposure to web-based media food marketing as an essential step in increasing the accountability of industries and governments in protecting children.	Data of the promotions of foods and beverages were collected through a video recording mobile device screen for two weekdays and one weekend day any time the participants went onto relevant web-based platforms or apps. For Android devices, an application was used (Lollipop screen recorder). The content and nutrient analysis of the promoted products was carried out using the World Health Organization European Region (WHO-EURO) Nutrient Profile Model (NPM).
Kidd et al. (2021) [30]	New Zealand	Facebook	Adolescents (16–18 years)	To test the feasibility of a browser extension to estimate the exposure of adolescents to (un)healthy food and beverage advertisements on Facebook and the persuasive techniques used to market these foods and beverages.	Data were collected through a browser extension (AdHealth) to automatically identify the type of advertisement seen and the duration of each ad sighting and perform a nutritional analysis of food and beverage advertisements based on the WHO-EURO NPM.
Tatlow-Golden & Boyland (2021) [48]	Philippines	Facebook and Instagram	Children and adolescents (5–17 years)	To describe the extent and nature of the marketing of unhealthy items in the Philippines. A second objective: to describe the nutrient content of food marketing on the social media platforms most popular with children in the Philippines.	A content analysis of relevant food and beverage brands for the under-18 audiences, considering the sale of the products, was carried out. Nutrient analysis was performed based on the WHO NPM for the Western Pacific Region.
Theodore et al. (2021) [31]	Mexico	Facebook, Twitter, and YouTube	Children and adolescents	To identify general characteristics, the use of persuasive techniques, and the nutritional quality of the Mexican digital marketing of food and beverage brands with the greatest number of followers and views (Facebook, Twitter, and YouTube) and with specific appeal to children/adolescents.	Multi-step process, including the identification of food and beverage/brand companies with the largest audience through social media analytics (Socialbakers) and nutritional quality according to the PAHO NPM.
Bragg et al. (2020) [25]	US	Instagram, Facebook, Twitter, Vine, and Tumblr	Youth	To identify the prevalence of social media advertising among fast food, beverage, and snack companies and examine the advertising techniques they use on Instagram, Facebook, Twitter, Tumblr, and Vine.	Identification of fast food, beverage, and snack brands with the highest advertising expenditures in the US across the top five most popular social media platforms among adolescents in 2014 to 2015. There must exist official US brand accounts. A content analysis to identify the marketing techniques of food and beverages was conducted.
Olstad and Lee (2020) [24]	Canada	NA	Children	NA. This is a commentary about the importance of including artificial intelligence to strengthen the ‘WHO CLICK’ methodology to monitor the marketing of unhealthy food on digital media.	Methodology proposal based on the CLICK monitoring framework with the addition of an artificial intelligence system to monitor unhealthy food and brand marketing to children on digital media.
Coates et al. (2019) [32]	UK	YouTube	Children (5–15 years)	To explore the extent and nature of food and beverage cues featured in YouTube videos of influencers popular with children.	Identification of food and beverage cues in the YouTube videos of two influencers. Content and nutrient analysis (using the UK NPM) was carried out.
Jaichuen et al. (2019) [33]	Thailand	Facebook	Children and adolescents	To assess the marketing of food on Facebook in relation to government regulations and the industry’s self-regulatory codes in Thailand.	Using the Socialbakers platform, the 30 most popular Facebook pages of food and beverage brands in the country were identified. A content analysis of marketing techniques was conducted, determining whether marketing strategies comply with Government regulations and the industry’s self-regulatory code.
Potvin et al. (2019) [34]	Canada	Facebook, Instagram, Snapchat, Twitter, and YouTube	Children (7–11 years) and adolescents (12–16 years)	To compare the frequency and healthfulness of food marketing seen by children and adolescents on social media apps as well as to estimate their weekly exposure.	Data on the promotions of foods and beverages were recorded through Tobii Pro Glasses. Content and nutrient analysis using the Pan-American Health Organization (PAHO) NPM and the UK NPM was carried out.
Brownbill et al. (2018) [35]	Australia	Facebook	Young adults (13–25 years)	To explore how sugar-sweetened beverages are marketed to Australian young people through sugar-sweetened beverage brand Facebook pages.	Content analysis of official posts from the six most popular Australian sugar-sweetened beverage brand Facebook pages (defined as those with a minimum of 50,000 unique adolescent visitors), which included information on ‘likes’, ‘comments’, ‘shares’, and ‘hashtags’.
Potvin et al. (2018) [36]	Canada	Websites	Adolescents (12–17 years)	To document the frequency and healthfulness of pop-up and banner food advertisements displayed on third-party websites preferred by adolescents in Canada.	The content analysis of the 10 most popular websites of food and beverages and the nutrient analysis were based on the PAHO and UK NPM. The identification of the websites was done through social media analytics (comScore).
Vandevijvere et al. (2018) [37]	New Zealand	Facebook and YouTube	Adolescents (13–18 years)	To analyze the extent, nature, and potential impact of marketing by food and beverage brands popular in New Zealand on Facebook and YouTube.	Identification of the most popular food and beverage brands on Facebook and YouTube through Socialbakers. Content analysis of the marketing techniques of popular food and beverages brands, including the type of post, the product type, and the total number of ‘likes’, ‘shares’, and ‘comments’.
WHO Regional Office for Europe (2018) [49]	WHO Europe member states ^1^	N/S	Children	To elucidate the rapidly changing digital marketing ecosystem within which action to protect children’s online experience must be taken. It then sets out two practical actions that can feasibly be undertaken: (1) the CLICK monitoring framework and (2) Proposed Policy Prerequisites.	This report proposed the five-step CLICK tool to monitor the extent to which children are exposed to the marketing of unhealthy products online. The five steps of the tool are: Comprehend the digital ecosystem (C); Landscape of campaigns (L); Investigate exposure (I); Capture on-screen (C), and Knowledge sharing (K).
Vandevijvere et al. (2017) [38]	New Zealand	Television, sports, magazines, and Facebook	Children and adolescents	To assess the extent and nature of unhealthy food marketing to New Zealand children and adolescents through the internet.	Content analysis of the marketing techniques of the most popular websites of food and beverages was conducted. The identification of the websites was done through social media analytics (AC Nielsen). Products were classified according to the New Zealand Ministry of Health Food and Beverage Classification.
Tatlow-Golden et al. (2016) [47]	Ireland	Facebook and company websites	Children and young people	To make essential first steps in identifying the digital food and drink marketing appealing to, or directed at, children and young people in Ireland.	Most popular brands of food and beverages were identified through the feature Create Adverts on Facebook. A content analysis was carried out.
Freeman et al. (2014) [39]	Australia	Facebook	Children and adolescents	To assess the amount, reach, and nature of energy-dense, nutrient-poor food and beverage marketing on Facebook.	Identification through Socialbakers of the 27 most popular Facebook pages of food and beverage brands. A content analysis of the marketing techniques used by the brands was conducted.
Kelly et al. (2013) [50]	Canada	Television and social media	Children	To identify approaches to monitoring food promotions via dominant media platforms responses by a review of studies measuring the nature and extent of exposure to food promotions.	Methodology proposal based on a multistep process according to the type of media (social media, television, among others).

Note: For more details, consult Appendix A. ^1^ Albania, Andorra, Armenia, Austria, Azerbaijan, Belarus, Belgium, Bosnia y Herzegovina, Bulgaria, Croatia, Cyprus, Czechia, Denmark, Estonia, Finland, France, Georgia, Germany, Greece, Hungary, Iceland, Ireland, Israel, Italy, Kazakhstan, Kyrgyzstan, Latvia, Lithuania, Luxembourg, Malta, Monaco, Montenegro, Netherlands, Nepal, North Macedonia, Norway, Poland, Portugal, Republic of Moldova, Romania, Russian Federation, San Marino, Serbia, Slovakia, Slovenia, Spain, Sweden, Switzerland, Tajikistan, Turkey, Turkmenistan, Ukraine, United Kingdom of Great Britain and Northern Ireland, and Uzbekistan. NA: Not applicable; BMS: breast-milk substitutes; The Code: International Code of Marketing of Breast-milk Substitutes; IBFAN: International Baby Food Action Network; NPM: Nutrient Profile Model; ICDC: International Code Documentation Centre; WHO: World Health Organization; IF: infant formula.

## Data Availability

The data presented in this study are available on reasonable request from the corresponding author.

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
