# Peer review of "Methodologies for Monitoring the Digital Marketing of Foods and Beverages Aimed at Infants, Children, and Adolescents (ICA): A Scoping Review"

_ijerph, 2022, doi:10.3390/ijerph19158951_

Round 1

Reviewer 1 Report

I like the idea embodied in the document, I think it can be an important pillar for future research and will be a reference for many other articles. I appreciate the effort and time devoted to improving knowledge on epidemiological findings in improving child and adolescent health through nutrition. However, I have some comments to strengthen your work.

General changes:

I consider that the document is very extensive, both in the results section and in the discussion, and certain ideas and information should be clarified. Excessively long sentences have been created that make it difficult to understand the text and are not common in English (Example: line 40-46)

Introduction

Line 57: Reformulate this sentence again because, in addition to being very long, it indicates twice "the aim"

Material and Methods

Line 118: It should not be said "in an attempt to" since the authors have not tried it, it is the methodology they have used.

Table 1 (Concept and population studied): How have the authors been able to discriminate to know that these resources were aimed at children under 19 years of age? Why 19 and not 18 or 16? Also, it would be important to better clarify the age of the target population since other different study groups have been identified (line: 211. Study population)

Line 132: all search strings used in the methodology must be described, not just some. However, in addition to search strings, other methods may be described as identified by authors.

Line 141, 142 y 143: It is not necessary to specifically mention the researchers who performed each task

Results

Line 153 and Figure 1: Do not duplicate information in text and figure. In the text it is not clear that the total resources for analysis are 420 and that 396 was excluded and the causes. If the figure is left (clarifies much more) identify causes of exclusion of the 396 ​​and eliminate one of the boxes where 420 resources appear.

Line 185: what country is this?

Line 192: it is a bit confusing because according to the text it seems that figure 3 only identifies the countries with a single resource but looking at the figure and the different shades of color, it seems that this is not the case. Clarify this.

Line 321-323: if it is the same work, put the review at the end of the text.

Line 618: It’s recommended that this tool “WHO CLICK” was enclosed in quotes.

I recommend the authors to re-edit the sections included in 3.5 (3.5.1, 3.5.2, 3.5.3, 3.5.4, 3.5.5, 3.5.6, 3.5.8, 3.5.9). This recommendation is focused on clarifying, organizing, and better classifying the findings found and therefore facilitating the reading of the article. It can be seen that the authors comment and describe types of resources ("Social Media Sites", "Websites consulted", "post", "videos"...) different types of methodologies, analysis and monitoring of information (questionnaires, search engines, WHO CLICK tool, the IBFAN-ICDC specific tool..). a

Line 601: Subsection 3.5.7 does not clarify why it has an ethnographic focus.

Discussion

It is important to clarify the study population and the different age groups. Since throughout the document there are different classifications. In addition, in the discussion section a news groups of digital exposure to this type of food "pregnant women" and “mothers of children 748 under 36 months” (line 736, 748, and 826) is also identified

The objective of this review was “identify and map the types of methodologies available to monitor BMS and FBHFSS digital marketing for ICA worldwide”, but both the results and the discussion mixed information. On the one hand, there would be the people involved in the consumption of this digital information (children and adolescents, parents, and pregnant women, but no infants), on the other hand, there would be the type of products (for infants-BMS and for children and adolescents-FBHFSS) and finally, the people or population groups that this type of digital marketing harms (pregnant women, mother of children under 36 months, infants, children, and adolescents under 19 years of age). Therefore, I insist that it is important that the authors consider re-editing the document. Firstly, to further synthesize the information and secondly to clarify these concepts.

Author Response

Dear reviewer, 

We are grateful for the opportunity to resubmit our manuscript “Methodologies for Monitoring the Digital Marketing of Foods and Beverages aimed at Infants, Children and Adolescents (ICA): A Scoping Review” (ijerph-1782552) to the International Journal of Environmental Research and Public Health. The comments and suggestions you provided us  were clear and insightful and they undoubtedly helped us to improve the manuscript. All changes in the manuscript are marked up using the “Track Changes” function. Attached a point-by-point discussion of all issues that were raised. 

Kind regards,

Reviewer 2 Report

Dear Editor and Authors,

Thank you for the opportunity to review the manuscript entitled “Methodologies for Monitoring the Digital Marketing of Foods 2 and Beverages aimed at Infants, Children and Adolescents 3 (ICA): A Scoping Review”

First of all, I want to congratulate you on your research efforts. Scoping reviews are particularly useful when the literature has not yet been comprehensively reviewed or exhibits a complex nature but sometimes it is a very difficult process.

Digital technologies are increasingly used for marketing food products throughout the world. The advent of social networking sites and other online communities presents new opportunities and challenges for the promotion, protection, and support of breastfeeding. Unfortunately, Digital Marketing also represents a major threat to the promotion of foods and beverages high in saturated fat, salt and/or free sugars(FBHFSS). Therefore, the authors' approach is a very interesting topic and worthy of publication.

The manuscript is clear, relevant for the field and presented in a well-structured manner with good research methodology. All figures and tables are appropriate but please consider putting Recommendations from figure 5 also into the text. Moreover, in-depth discussions are required,  especially, they should be more critical. (Which Types of Methodologies for Monitoring Digital Marketing are the most effective?). In addition, the final chapter can also be supplemented with directions for further research.

Overall, I enjoyed reading this paper. The topic is both interesting and very important.

Author Response

(The authors gave the same response as above.)

Reviewer 3 Report

This paper provides a comprehensive review of the research conducted on unethical marketing practices of supposedly unhealthy foods that are breast-milk substitutes and foods and beverages that are high in saturated fat, salts, and/or free sugar, namely BMS and FBHFSS. This paper summarizes well the methodologies and research findings in this domain. You said digital marketing of baby formula and processed meat challenges human rights on pp.26. This is a strong and rather irrelevant statement. My main concern with this paper is that the authors might need to adopt a neutral tone as infant formula is not intrinsically unhealthy and mix feeding is not considered an unhealthier practice than exclusive breastfeeding. Overall, this paper has substantial merits to the general readership, particularly the research community in this domain.

Author Response

(The authors gave the same response as above.)

Round 2

Reviewer 1 Report

I consider that the work could have been further synthesized to make it easier to read and understand.

However, the authors have implemented the proposed changes to improve their final result.